# The read-through transcription-mediated autoactivation circuit for virulence regulator expression drives robust type III secretion system 2 expression in *Vibrio parahaemolyticus*

**Dhira Saraswati Anggramukti[1]**, **Eiji Ishii[1,2]**, **Andre Pratama[1]**, **Mohamad Al Kadi[3]**, **Tetsuya Iida[1,2]**, **Toshio Kodama[4]**, **Shigeaki Matsuda[1,2]** *

1 Department of Bacterial Infections, Research Institute for Microbial Diseases, Osaka University, Osaka, Japan, 2 Center for Infectious Disease Education and Research, Osaka University, Osaka, Japan, 3 Human Immunology (Single Cell Genomics), WPI Immunology Frontier Research Center, Osaka University, Osaka, Japan, 4 Department of Bacteriology, Institute of Tropical Medicine, Nagasaki University, Nagasaki, Japan

☯ These authors contributed equally to this work.
* matsudas@biken.osaka-u.ac.jp

## Abstract

*Vibrio parahaemolyticus* is the leading cause of seafood-borne gastroenteritis in humans worldwide. The major virulence factor responsible for the enteropathogenicity of this pathogen is type III secretion system 2 (T3SS2), which is encoded on the 80-kb *V. parahaemolyticus* pathogenicity island (Vp-PAI), the gene expression of which is governed by the OmpR-family transcriptional regulator VtrB. Here, we found a positive autoregulatory feature of *vtrB* transcription, which is often observed with transcriptional regulators of bacteria, but the regulation was not canonically dependent on its own promoter. Instead, this autoactivation was induced by heterogeneous transcripts derived from the VtrB-regulated operon upstream of *vtrB*. VtrB-activated transcription overcame the intrinsic terminator downstream of the operon, resulting in transcription read-through with read-in transcription of the *vtrB* gene and thus completing the autoregulatory loop for *vtrB* gene expression. The dampening of read-through transcription with an exogenous strong terminator reduced *vtrB* gene expression. Furthermore, a *V. parahaemolyticus* mutant with defects in the *vtrB* autoregulatory loop also showed compromises in T3SS2 expression and T3SS2-dependent cytotoxicity *in vitro* and enterotoxicity *in vivo*, indicating that this autoregulatory loop is essential for sustained *vtrB* activation and the consequent robust expression of T3SS2 genes for pathogenicity. Taken together, these findings demonstrate that the regulatory loop for *vtrB* gene expression based on read-through transcription from the upstream operon is a crucial pathway in T3SS2 gene regulatory network to ensure T3SS2-mediated virulence of *V. parahaemolyticus*.

**Data Availability Statement:** All data are available in the article and supporting information.

**Funding:** This study was supported by Grants-in-Aid for Scientific Research from the Japan Society for the Promotion of Science (Grants 20K07428 and 23K06529 to S.M. and Grants 20K15748 and 23K05637 to E.I.), the Institute for Fermentation (to S.M.), the Chemo-Sero-Therapeutic Research Institute (to S.M.), the Joint Usage / Research Center on Tropical Disease, Institute of Tropical Medicine, Nagasaki University (2023-Ippan-14 to S.M.), the Center for Infectious Disease Education and Research (to S.M. and E.I.), and by the Taniguchi scholarship program from BIKEN Foundation (to D.S.A.). The funders had no role in study design, data collection, and interpretation, or the decision to submit the work for publication.

**Competing interests:** The authors have declared that no competing interests exist.

## Author summary

Many bacterial transcription factors undergo autoregulation, a process by which transcription factors regulate their own transcription to amplify or reduce the output responses to a changing environment. The common mode of such autoregulation is achieved by the transcription factor acting on its own promoter. In this study, we found that VtrB, a virulence regulator of the T3SS2 genes of the major food-borne pathogen *Vibrio parahaemolyticus*, autoregulates its own expression but independently of its own promoter. We also demonstrated how this autoregulation occurs: VtrB activates transcription of the upstream operon and transcription extends to the *vtrB* gene over the relatively less effective intrinsic terminator. Read-through transcription thus underlies the autoregulatory loop of *vtrB* expression. Moreover, this autoregulation was essential for the amplification of *V. parahaemolyticus* T3SS2 expression and induction of the full virulence of this pathogen. Together, our findings not only offer new insights into how *V. parahaemolyticus* controls its virulence gene expression to ensure pathogenicity but also provide a framework for further exploring the analogous mechanisms for the autoactivation of transcription factor gene expression in bacteria.

## Introduction

*Vibrio parahaemolyticus* is a gram-negative halophilic bacterium that causes acute gastroenteritis in humans, and infections with this pathogen have been spreading on a global scale over the last quarter century [1–3]. This bacterium inhabits marine and estuarine environments, and its infections are associated with the consumption of raw or undercooked seafood. The major virulence factor responsible for the enteropathogenicity of *V. parahaemolyticus* is one of the two type III secretion systems, called T3SS2 [4,5]. The T3SS is a multicomponent syringe-like protein secretion apparatus that is widespread in gram-negative pathogens and symbionts and injects bacterial proteins, so-called effectors, directly into target eukaryotic host cells, which results in disturbing the functions of host cells and promoting infections or symbiosis [6–8]. The T3SS2-related genes are encoded into an 80-kb pathogenicity island called Vp-PAI, which is located on the second of two chromosomes in *V. parahaemolyticus* [4]. The Vp-PAI region contains putative mobile elements and a lower guanine-cytosine content compared with that of the whole genome, which serve as indicators of an exogenous DNA region [4,9], and this region is thought to have been acquired by horizontal gene transfer, an event that is putatively mediated by Tn7-CRISPR [10].

The expression of the T3SS2 gene cluster in the Vp-PAI region is regulated by two OmpR-family transcription factors, *V. parahaemolyticus* T3SS2 regulator A (VtrA) and *V. parahaemolyticus* T3SS2 regulator B (VtrB), both of which are also encoded within the Vp-PAI region, in a cascade manner [11]. VtrA is a single membrane-spanning protein with an N-terminal OmpR-family DNA-binding domain and a C-terminal periplasmic domain that forms a complex with its cotranscribed protein VtrC [12]. VtrA directly activates the gene expression of VtrB through binding to the area around the -35 element within the *vtrB* promoter region [13]. The activation of *vtrB* transcription also requires another membrane-spanning regulator, ToxR [14], which also binds upstream of the VtrA-binding region in the *vtrB* promoter [13]. VtrB is also a membrane-localized regulator protein with an N-terminal OmpR-family DNA-binding domain and a C-terminal transmembrane domain that promotes the expression of T3SS2-related genes [11]. *vtrB* gene expression is altered by changes in physical conditions such as temperature and salinity, and this alternation is mediated by the histone-like nucleoid-

structuring protein through binding upstream of the *vtrB* gene [13]. Bile acids in bile secreted into the intestinal tract of the host activate *vtrB* gene expression [15], which promotes VtrA oligomerization, and thus facilitates VtrA binding to the *vtrB* promoter and subsequent transcriptional activation of *vtrB* [16]. Such expression profiles appear to reflect the adaptation of *V. parahaemolyticus* to the human body environment during infection, in which transcriptional activation of *vtrB* is thus crucial.

Bacteria have complex gene regulatory networks to enable appropriate output responses to changes in the environment. One common motif of such a regulatory network is autoregulation, a process by which transcription factors regulate their own transcription, either directly or indirectly, and positively or negatively [17,18]. Indeed, many bacterial transcription factors undergo autoregulation, such as in *E. coli*, in which half of the known transcription factors are self-regulated [19]. Positive autoregulation generally stimulates more production of the transcription factor and thereby amplifies the gene expression response regulated by the transcription factor. In *V. parahaemolyticus*, the regulatory cascade through which VtrA activates *vtrB* transcription has been well described, but the regulatory network involving the transcriptional activation of *vtrB* and downstream T3SS2 genes remains to be characterized. Here, we investigated whether *vtrB* transcription is autoregulated to refine its own expression. We found that *vtrB* gene expression is upregulated in an autoregulatory manner, but not by acting on its own promoter. Instead, this autoactivation is induced by transcription from the operon upstream of *vtrB* through read-through transcription across the intrinsic terminator. Furthermore, this positive autoregulation allows an increase in the expression level of VtrB and thereby ensures the expression of T3SS2 genes in *V. parahaemolyticus* and hence the full virulence of this pathogen.

## Results

### *vtrB* gene expression has an autoactivating feature

To examine the autoregulatory property of *vtrB* transcription in *V. parahaemolyticus*, we compared *vtrB* gene expression between the wild type (WT) and the *vtrB*-deleted (Δ*vtrB*) strains. To this end, *V. parahaemolyticus* strains were grown in LB medium with 0.3 M NaCl to inactivate *vtrB* gene expression [13] (henceforth referred to as the nonpermissive condition) or in LB medium with 0.3 M NaCl and 80 μM sodium taurodeoxycholate (TDC) to activate *vtrB* gene expression [15] (henceforth referred to as the inductive condition). The *vtrB* gene expression in each strain was measured by quantitative real-time polymerase chain reaction (qRT–PCR) targeting the 5'-untranslated region of the *vtrB* gene (*vtrB* 5'-UTR), which corresponds to the 102-bp region starting from the +1 transcription start site (TSS) of the *vtrB* gene [16] (Fig 1A). Consistent with the previous finding that VtrA activates *vtrB* transcription in a bile acid-dependent manner [15,16], *vtrB* 5'-UTR expression was activated in the WT strain under the inductive condition but was abolished in the *vtrA*-deleted strain (Δ*vtrA*), supporting the validity of our assay. However, the *vtrB* 5'-UTR transcript was also decreased in the Δ*vtrB* strain compared with the WT strain (Fig 1B), and this decrease was restored by expression of plasmid-borne VtrB in the *vtrB*-deleted strain (Δ*vtrB*) (Fig 1C). Furthermore, the plasmid-borne expression of VtrB could confer *vtrB* 5'-UTR expression in the *vtrA*- and *vtrB*-deleted strain (Δ*vtrA* Δ*vtrB*), which lack *vtrB* gene activation by VtrA [11] (Fig 1C). Taken together, these results revealed the autoactivating feature of *vtrB* gene expression.

The autoactivation of transcription factor genes is often achieved by acting on their own promoters [17,18]. Therefore, to determine whether VtrB acts on its own promoter to activate *vtrB* transcription, we employed a pHRP309-derived *lacZ* transcriptional fusion reporter plasmid containing a 284-bp upstream promoter region of *vtrB* (P*vtrB*), pHRP309-P*vtrB* [16], to

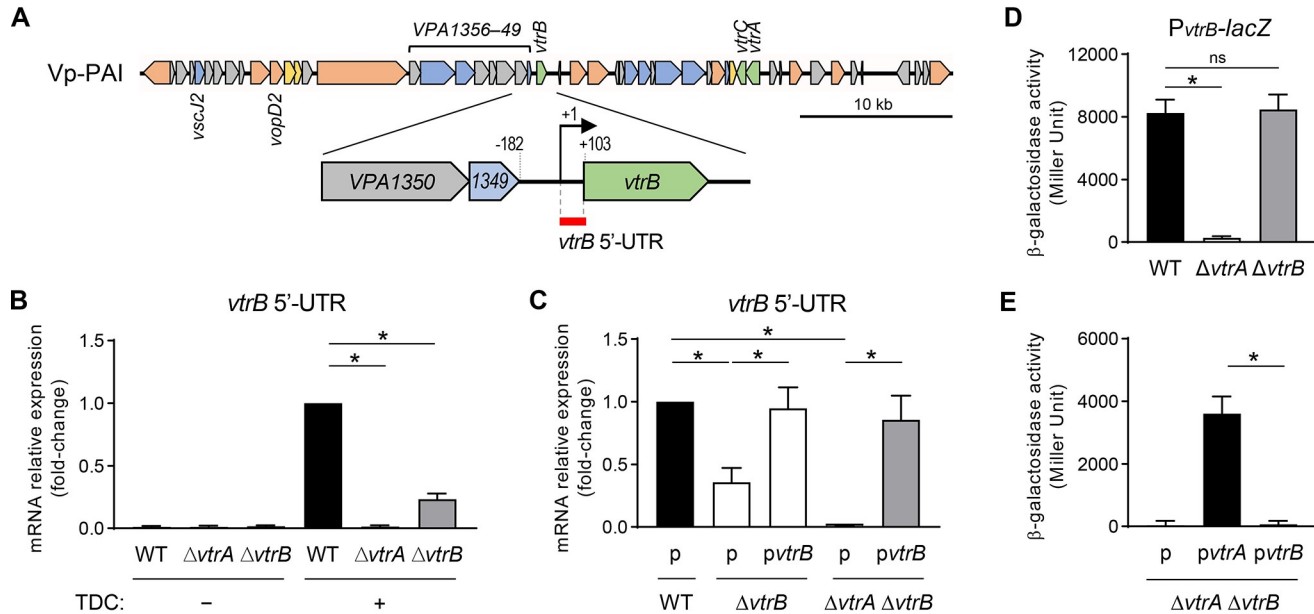

**Fig 1. *vtrB* gene expression exhibits an autoactivating feature.** (**A**) Schematic representation of the Vp-PAI region (*VPA1370–VPA1321*: top) and the *vtrB* upstream region with adjacent genes (bottom). The arrows indicate open reading frames (ORFs) and their orientation. The ORFs are colored according to their function: T3SS2 apparatus genes, blue; genes encoding T3SS2-secreted proteins, orange; other T3SS2-associated proteins, yellow; regulator genes, leaf green; hypothetical genes, gray. A long operon upstream of the *vtrB* gene predicted by Operon-mapper (https://biocomputo.ibt.unam.mx/operon_mapper/) [20] is indicated above the arrows. The region targeted by the qRT–PCR primers used in **B** and **C** is indicated by a red line as *vtrB* 5'-UTR. (**B**) Relative expression of the *vtrB* 5'-UTR region in *V. parahaemolyticus* WT, Δ*vtrA*, and Δ*vtrB* strains grown in LB medium containing 0.3 M NaCl at 37˚C with or without TDC. Total RNA was extracted from each culture at an $OD_{600}$ of 1 and was analyzed by qRT–PCR. The mean fold-change and standard deviation (SD) values are indicated relative to the WT strains (n = 3). *, $p < 0.05$, as revealed by one-way ANOVA followed by Dunnett's multiple comparison test. (**C**) Effect of *vtrB* complementation under the control of the $P_{BAD}$ promoter on *vtrB* 5'-UTR expression. *V. parahaemolyticus* WT, Δ*vtrB* and Δ*vtrA*Δ*vtrB* strains with pBAD18-Cm empty vector (indicated as p) or pBAD18-Cm-*vtrB* (indicated as p*vtrB*) were grown under TDC and arabinose induction. The mean and SD values are indicated relative to the WT strain harboring the empty vector (n = 3). *, $p < 0.05$, as indicated by one-way ANOVA followed by Dunnett's multiple comparison test. (**D**) β-galactosidase activity from the *lacZ* fusion reporter of the *vtrB* promoter region ($P_{vtrB}$) in *V. parahaemolyticus* WT, Δ*vtrA*, and Δ*vtrB* strains grown under TDC induction. The values show the means, and the error bars represent S.D. (n = 3). *, $p < 0.05$; ns, not significant, as revealed by one-way ANOVA followed by Dunnett's multiple comparison test. (**E**) β-galactosidase activity from the $P_{vtrB}$-*lacZ* reporter of the *V. parahaemolyticus* Δ*vtrA*Δ*vtrB* strain with pBAD18-Cm empty vector (indicated as p), pBAD18-Cm-*vtrA* (indicated as p*vtrA*), or pBAD18-Cm-*vtrB* (indicated as p*vtrB*) was grown under TDC and arabinose induction. The values show the means, and the error bars represent S.D. (n = 3). *, $p < 0.05$, as revealed by Student's t test.

monitor *vtrB* promoter activity. pHRP309-$P_{vtrB}$ was introduced into *V. parahaemolyticus* strains, and $P_{vtrB}$-*lacZ* expression was assessed by measuring β-galactosidase activity under the inductive condition (Fig 1D and 1E). The $P_{vtrB}$-*lacZ* expression was not observed in the Δ*vtrA* strain, confirming that VtrA is responsible for the activation of this promoter, while the Δ*vtrB* strain retained $P_{vtrB}$-*lacZ* activity at a level that was comparable to the WT strain (Fig 1D). Moreover, the *vtrB* promoter was inactive in the Δ*vtrA* Δ*vtrB* strain harboring the empty vector but was activated in the Δ*vtrA* Δ*vtrB* strain expressing VtrA on the plasmid, whereas the plasmid-borne expression of VtrB did not confer *vtrB* promoter activity to the Δ*vtrA* Δ*vtrB* strain (Fig 1E). Taken together, these results suggest that the *vtrB* promoter is not activated by VtrB itself.

## Multiple transcripts of different lengths contain *vtrB* gene

To address the mechanism by which *vtrB* autoactivation is driven independently of its own promoter, we profiled the *vtrB* transcript in *V. parahaemolyticus* strains grown under the inductive condition by northern blotting using a probe against the *vtrB* promoter region (Fig 2A). Remarkably, two major *vtrB* transcripts of different lengths were detected in the WT

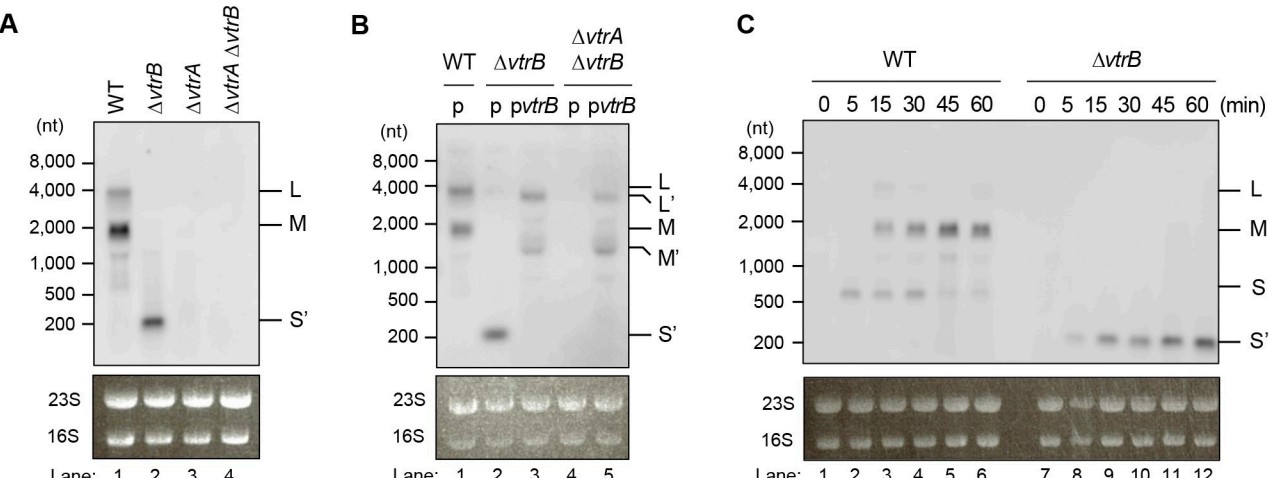

**Fig 2. *vtrB* is transcribed as multiple transcripts of different lengths.** (**A**) *vtrB* transcript profile in *V. parahaemolyticus* WT, Δ*vtrB*, and Δ*vtrA* Δ*vtrB* strains grown under TDC induction. Total RNA was extracted once the cells reached an $OD_{600}$ of 1, and northern blotting was performed using the $P_{vtrB}$ probe. L, 4,000-nt L transcript; M, 2,000-nt M transcript; S', S transcript after *vtrB* deletion. 23S rRNA and 16S rRNA served as loading controls. (**B**) *vtrB* expression affects the *vtrB* transcript profile. *V. parahaemolyticus* WT, Δ*vtrB*, and Δ*vtrA* Δ*vtrB* strains with pBAD18-Cm empty vector (indicated as p) or pBAD18-Cm-*vtrB* (indicated as p*vtrB*) were grown in the presence of TDC and 0.1% arabinose, and total RNA was extracted once the cells reached an $OD_{600}$ of 1. Northern blotting was performed using the $P_{vtrB}$ probe. L, 4,000-nt L transcript; L', L transcript after *vtrB* deletion; M, 2,000-nt M transcript; M', M transcript after *vtrB* deletion; S', S transcript after *vtrB* deletion. (**C**) Time-course analysis of the *vtrB* transcript profile in the *V. parahaemolyticus* WT and Δ*vtrB* strains. Both strains were grown to an $OD_{600}$ of 0.8, and TDC was then added. RNA was extracted after 0, 5, 15, 30, 45, and 60 min of TDC induction, and northern blotting was performed using the $P_{vtrB}$ probe. L, 4,000-nt L transcript; M, 2,000-nt M transcript; S, 700-nt S trasncript; S', S transcript after *vtrB* deletion. The data are representative of three independent experiments (**A–C**).

strain (Fig 2A: lane 1). Based on their migration relative to the RNA markers, the estimated sizes of the long and short transcripts were approximately 4,000 nt and 2,000 nt, respectively, which cannot be attributed to VtrA-dependent transcription from the *vtrB* promoter because the coding sequence of *vtrB* has a length of 552 bp and the previously mapped TSS of the *vtrB* gene from the *vtrB* promoter region is located 102 bases upstream of the start codon of the *vtrB* coding sequence (S1A Fig) [16]. In contrast, a smaller *vtrB* transcript of approximately 200 nt was detected in the Δ*vtrB* strain (S1B Fig: a 410-bp deletion in the *vtrB* coding sequence) [11] (Fig 2A: lane 2) and was hypothesized to be a transcript from the *vtrB* promoter based on its length. No *vtrB* transcript was observed in the Δ*vtrA* strain and the Δ*vtrA* Δ*vtrB* strain (both of which lack *vtrB* transcription from the *vtrB* promoter) (Fig 2A: lanes 3 and 4), supporting the notion that these *vtrB* transcripts are generated in a VtrA-dependent manner. The expression of VtrB on the plasmid in the Δ*vtrB* strain yielded two high-molecular-weight transcripts (~3,600 nt and ~1,600 nt), which corresponded to the lengths of two transcripts observed in the WT strain minus the length of the *vtrB* gene deletion, but the smaller transcript of ~200 nt was not observed (Fig 2B: lane 3). A similar transcript profile was found in the Δ*vtrA* Δ*vtrB* strain with plasmid-borne expression of VtrB (Fig 2B: lane 5). To further investigate the temporal expression patterns of these transcripts, we performed a time-course analysis in which RNA samples were extracted at multiple time points after TDC induction (Fig 2C). In the WT strain, an ~700-nt transcript putatively derived from the *vtrB* promoter appeared 5 min after TDC induction, and high-molecular-weight transcripts (~4,000 nt and ~2,000 nt) were detected starting 15 min after TDC induction (Fig 2C: lanes 1–3). Thus, three different transcript sizes were observed after TDC induction in the WT strain. Herein, the ~4,000-nt, ~2,000-nt, and ~700-nt transcripts are referred to as L, M, and S transcripts, respectively. The levels of the L and S transcripts were decreased 30 min after TDC induction, whereas the M transcript showed sustained expression up to 60 min after TDC induction (Fig 2C: lanes 4–6),

which explains the transcript pattern of the WT strain shown in Fig 2A. In the Δ*vtrB* strain, the ~200-nt transcript containing a deletion in the *vtrB* coding sequence was consistently detected until 60 min after TDC induction (Fig 2C: lanes 7–12).

To further characterize the multiple species of *vtrB* transcripts, we mapped the 5'-end of the transcripts in the WT strain 15 min after TDC induction by rapid amplification of cDNA ends (RACE) using a gene-specific primer to amplify the transcripts from the 3'-region of the *vtrB* gene. Consistent with the lengths of the three *vtrB* transcripts observed by northern blotting, three 5'-RACE products of different sizes were observed by agarose gel electrophoresis (S2A Fig), and from these, four 5'-ends were determined (S2B Fig). The 5'-end of the smaller RACE product, which was expected to correspond to the S transcript, was mapped to 102 bases upstream of the start codon of the *vtrB* gene, which is consistent with the previously mapped TSS of the *vtrB* gene (S2C Fig: top). Moreover, the 5'-end of the larger RACE product, which was expected to correspond to the M transcript, was mapped to two sites, and both of these sites were located approximately 1,200 bp upstream of the start codon of the *vtrB* gene, which is upstream of *VPA1350* and within the coding sequence of *VPA1351* (S2C Fig: middle). The 5'-end of the largest RACE product, which was expected to correspond to the L transcript, was mapped to 39 bases upstream of the *VPA1353* start codon, which is within the coding sequence of *VPA1354* (S2C Fig: bottom). Together, these results revealed multiple species of *vtrB* transcripts, and among these, the L and M transcripts could not be induced by transcription from the *vtrB* promoter.

## The long transcripts originate by transcription from the promoter of the *VPA1356–VPA1349* operon upstream of the *vtrB* gene

Upstream of the *vtrB* gene, a predicted operon encompasses the *VPA1356* to *VPA1349* genes, and the expression of these genes is dependent on VtrB [11]. Because the 5'-ends of the L and M transcripts were mapped to the interior of this operon, we first hypothesized that the internal transcription start induced the transcripts. To test this possibility, the transcription from upstream of the 5'-ends of the L and M transcripts was examined. The 318-bp upstream region of the *VPA1353* gene or the 304-bp upstream region of the *VPA1350* gene (Fig 3A) was inserted upstream of the promoterless *lacZ* gene on the pHRP309 plasmid, yielding the UP$_{VPA1353}$-*lacZ* or UP$_{VPA1350}$-*lacZ* transcriptional fusion reporter to assess the transcriptional activation of this region in *V. parahaemolyticus*. However, both the UP$_{VPA1353}$-*lacZ* and UP$_{VPA1350}$-*lacZ* reporters were inactive in the WT strain (Fig 3B: Fragments 2 and 3), suggesting the absence of an internal TSS around the 5'-end of the L and M transcripts within these regions. To seek the TSSs of the L and M transcripts, we prepared a series of *lacZ* reporter plasmids containing the further upstream region of the *vtrB* gene (Fig 3A). The expression of UP$_{VPA1356}$-*lacZ*, the *lacZ* gene fused to the upstream region of the *VPA1356* gene at the start of the predicted operon, was activated in the WT but not the Δ*vtrB* strain (Fig 3B: Fragment 5), supporting the existence of VtrB-activated transcription from upstream of the start of the operon. VtrB-dependent *lacZ* activity was similarly observed with the UP$_{VPA1356}$-UP$_{VPA1350}$-*lacZ*, UP$_{VPA1356}$-*VPA1349*-*lacZ*, and UP$_{VPA1356}$-DN$_{VPA1349}$-*lacZ* reporters (Fig 3B: Fragments 6, 7, and 8), whereas the *VPA1356*-UP$_{VPA1350}$-*lacZ* reporter showed no *lacZ* activity (Fig 3B: Fragment 4), suggesting that the transcription of this region is dependent on the single *VPA1356* promoter at the start of the operon. The UP$_{VPA1356}$-P$_{vtrB}$-*lacZ* reporter containing both the *VPA1356* and *vtrB* promoters was also activated at a higher level than the reporters containing either the *VPA1356* or *vtrB* promoters, but it should be noted that the *lacZ* activity obtained with the UP$_{VPA1356}$-P$_{vtrB}$-*lacZ* reporter under this condition was lower than the cumulated activities of the UP$_{VPA1356}$-*lacZ* and P$_{vtrB}$-*lacZ* reporters (Fig 3B: Fragments 1, 5

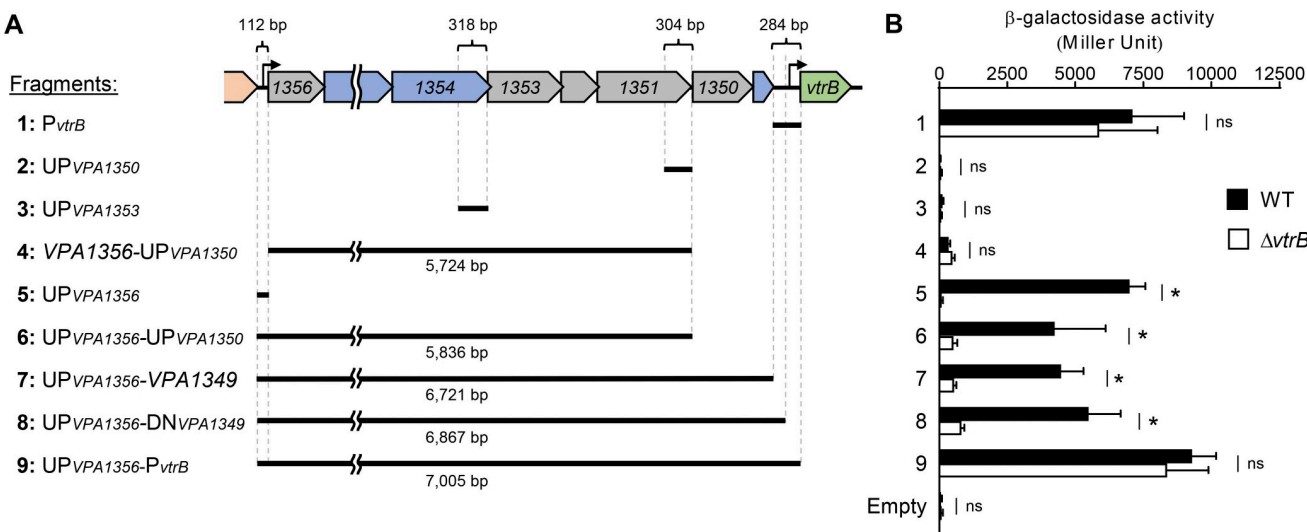

**Fig 3. VtrB activates its own expression from the upstream promoter of the *VPA1356–VPA1349* operon. (A)** Schematic representation of *lacZ* fusion reporters containing *vtrB* upstream regions of various lengths. The lengths of the long fragments are shown at the bottom of each fragment schematic. (**B**) β-galactosidase activity of the various constructs of *lacZ* fusion reporters in *V. parahaemolyticus* WT and Δ*vtrB* strains. The values show the means, and the error bars represent SDs. (n≥3). *, p < 0.05; ns, not significant, as revealed by Student's t test.

and 9). Taken together, these results suggest that the transcription initiation of the L and M transcripts depends on the promoter of the *VPA1356–VPA1349* operon. The inconsistent lengths of the L and M transcripts (~4,000 and ~2,000 nt) with an ~7.5-kb distance from *VPA1356* to *vtrB* indicated that these transcripts were generated from the *VPA1356–vtrB* transcript through some modification.

## Read-through transcription results from incomplete transcription termination at the intrinsic terminator of *VPA1349*

The transcription of the operon is generally terminated at a terminator positioned at the end of the operon. Among two well-known types of terminators, Rho-dependent and Rho-independent (intrinsic) terminators [21], we indeed predicted the presence of a typical intrinsic terminator with a 13-nt stem–loop and a 4-nt U-tract at the end of the *VPA1356–VPA1349* operon as the *VPA1349* terminator (*VPA1349*T) using ARNold (http://rssf.i2bc.paris-saclay.fr/toolbox/arnold/) [22] and Mfold (http://www.unafold.org/mfold/applications/rna-folding-form.php) [23] (S3A Fig, *VPA1349*T). Therefore, our observation that the L and M transcripts spanned across the predicted *VPA1349*T to the *vtrB* gene (Figs 2 and 3) indicates read-through transcription over *VPA1349*T. To validate the transcription termination or read-through at the *VPA1349* terminator, we performed northern blotting using probes against the *VPA1350* and *VPA1349* coding regions. Multiple transcripts in the WT but not Δ*vtrB* strain were detected on the blots with both probes, and among these, the longer RNAs with sizes of ~4,000 nt and ~2,000 nt were expected to be identical to the L and M transcripts, respectively (Fig 4A and 4B), as observed with the probe against the *vtrB* promoter region (Fig 4C). Additional transcripts of ~3,000 nt and approximately 1,000 nt were also detected, which is consistent with the lengths expected when the transcription of the L and M transcripts is terminated at *VPA1349*T. A doublet of approximately 1,000 nt was presumably due to two 5' ends at a distance of 122 bp from the M transcript mapped in the 5'-RACE (S2 Fig). Thus, these results indicate that transcription of the *VPA1356–VPA1349* operon is indeed terminated at

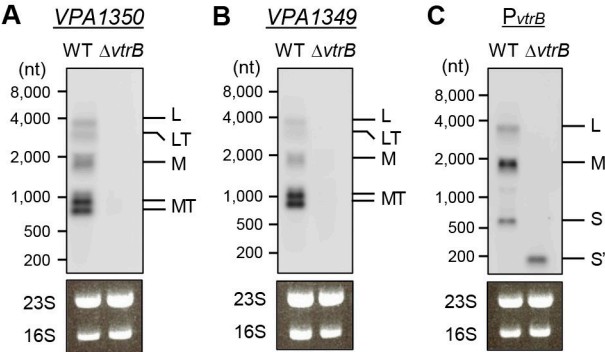

**Fig 4. Incomplete transcription termination at the *VPA1349* terminator (*VPA1349*T).** (**A–C**) *V. parahaemolyticus* WT and Δ*vtrB* strains were grown in LB medium containing 0.3 M NaCl to an OD$_{600}$ of 0.8, and TDC was then added. After further incubation for 15 min, total RNA was extracted, and northern blotting was performed using a probe for the *VPA1350* coding region (**A**), a probe for the *VPA1349* coding region (**B**), and the P$_{vtrB}$ probe (**C**). L, L transcript; LT, L transcript terminated at *VPA1349*T; M, M transcript; MT, M transcript terminated at *VPA1349*T; S, S transcript; S', S transcript after the *vtrB* deletion. 23S rRNA and 16S rRNA were used as loading controls.

*VPA1349*T, but a portion of the transcription reads through the terminator and extends to the coding sequence of the *vtrB* gene.

To examine the termination efficiency of *VPA1349*T, the terminator region was cloned and inserted into the pBAD-*lacZ* plasmid (pBAD-*VPA1349*T-*lacZ*). For comparison, we generated pBAD-*lacZ* carrying the termination-deficient mutant, in which the nucleotide substitution (GGGGC > CCCCG) was introduced to disrupt hairpin structure formation in *VPA1349*T (HP mutant, pBAD-*VPA1349*T-HP-*lacZ*) or pBAD-*lacZ* carrying the *rplL* terminator (*rplL*T) of *E. coli*, which is an established strong Rho-independent terminator [24] (S3A Fig: *rplL*T) (pBAD-*rplL*T-*lacZ*). These constructs were introduced into the *V. parahaemolyticus* Δ*vtrB* strain, and the effect of the terminator on *lacZ* expression was assessed by measuring β-galactosidase activity under the inductive condition. *VPA1349*T exhibited ~75% less β-galactosidase activity than the HP mutant, which showed moderate β-galactosidase activity (S3B Fig). In contrast, *rplL*T showed almost no β-galactosidase activity. Similar results were observed under the nonpermissive condition (S3B Fig), indicating that the transcription termination was not affected by TDC induction. The intrinsic terminator consists of a hairpin structure with a stem of GC-rich bases and an internal loop, which is followed by a sequence of U-rich tracts (S3C Fig) [25], and the strength of the terminator is generally associated with its thermodynamic stability as indicated by Gibbs free energy (ΔG), the higher this value is, the more difficult it is for the sequence to form a stable secondary structure [21,26,27]. We predicted the free energy of *VPA1349*T according to each element of the terminator's secondary structure: the stem structure (ΔG$_S$), the U-tract structure (ΔG$_U$), the loop structure (ΔG$_L$), and the total free energy (ΔG$_T$), using the Mfold program [23] (S1 Table) and compared them with those of *rplL*T. The computed ΔG$_U$ of *VPA1349*T was identical to that of *rplL*T (both ΔG$_U$ = −1.2 kcal/mol) because they contain almost a similar number of U-rich nucleotides following the stem–loop structure, which suggests that the U-rich tract may not contribute to the discrepancy between the termination abilities of *VPA1349*T and *rplL*T. Nevertheless, the other thermodynamic parameters support the notion that *VPA1349*T is markedly less stable terminator than *rplL*T: *VPA1349*T has higher ΔG$_T$, ΔG$_S$, and ΔG$_L$ values (−8.80 kcal/mol, −13.30 kcal/mol, and 5.70 kcal/mol, respectively) than *rplL*T (−21.60 kcal/mol, −25.20 kcal/mol, and 4.80 kcal/mol, respectively). Taken together, these results suggest incomplete transcription termination by *VPA1349*T, presumably due to the moderate thermodynamic stability of the hairpin structure.

## Read-through transcription of *vtrB* results in robust activation of the T3SS2 genes

We then determined whether read-through transcription mediates *vtrB* autoactivation and subsequent T3SS2 gene activation. To this end, we used a double terminator system [28,29] in which *rplL*T was placed downstream near the original intrinsic terminator to block read-through transcription at *VPA1349*T (S4A Fig). To first validate the system, we constructed pBAD-*lacZ* harboring the *VPA1349* coding region and *VPA1349*T (pBAD-*VPA1349*-T-*lacZ*) or harboring an additional *rplL*T adjacent to *VPA1349*T in pBAD-*VPA1349*-T-*lacZ* (pBAD-*VPA1349*-DT-*lacZ*). Both constructs were introduced into the *V. parahaemolyticus* Δ*vtrB* strain, and read-through transcription of the *lacZ* gene was assessed by measuring β-galactosidase activity under inductive and nonpermissive conditions. The strain carrying the pBAD-*VPA1349*-DT-*lacZ* plasmid exhibited ~60% less β-galactosidase activity than the strain with the pBAD-*VPA1349*-T-*lacZ* plasmid under both conditions (S4B Fig), validating the efficient termination of transcription by the double terminator system.

We then constructed a *V. parahaemolyticus* strain carrying the double terminator (DT strain), in which *rplL*T was inserted downstream of *VPA1349*T on the chromosome of the WT or POR-2 strain (TDH- and T3SS1-defective *V. parahaemolyticus*) [30]. In the WT-derived DT strain, an ~700-nt *vtrB* transcript was consistently detected 5–60 min after TDC induction, whereas multiple *vtrB* transcripts were observed in the WT strain (S4C Fig). We also observed ~3,000-nt, ~1,000-nt, and ~900-nt transcripts that were terminated at *VPA1349*T but not ~4,000-nt and ~2,000-nt read-through transcripts on the northern blots with *VPA1349* and *VPA1350* probes (S4D Fig), which was different from the results found for the WT strain. The qRT–PCR analysis also indicated that the DT strain exhibited defects in the expression of the intergenic region between *VPA1349* and the *vtrB* promoter (downstream of *VPA1349*), which differed from the results found for its parental POR-2 strain (Fig 5A), validating that read-through transcription over *VPA1349*T was indeed diminished by the double terminator on the chromosome. The DT strain showed impairments in the expression of *vtrB*, and in that of the VtrB-regulated T3SS2 genes *vopD2* (encoding a T3SS2-secreted protein), *vscJ2* (encoding a T3SS2 IM-ring apparatus protein), and *VPA1349* (located upstream of the double terminators) compared with the POR-2 strain. To further examine the effect of read-through transcription of *vtrB* on the T3SS2 gene activation, the protein production of T3SS2-associated proteins was determined by immunoblotting. In agreement with the qRT–PCR results, the protein levels of VtrB, VopD2, and VscJ2 in the DT strain were lower than those in the parental POR-2 strain (Fig 5B). The T3SS2-mediated secretion of VopD2 into the culture medium from the DT strain was also reduced compared with that from the POR-2 strain, indicating that the DT strain showed compromised T3SS2 secretion activity. Thus, these results revealed that read-through transcription increases *vtrB* expression, leading to subsequent T3SS2 gene activation.

## Read-through transcription of *vtrB* ensures the pathogenicity of *V. parahaemolyticus*

Given the crucial role of T3SS2 in the pathogenicity of *V. parahaemolyticus*, we wondered whether T3SS2 expression induced by *vtrB* autoactivation contributes to *V. parahaemolyticus* pathogenicity. T3SS2 causes cytotoxicity against some cultured cell lines, which is one of the T3SS2-mediated virulence traits of *V. parahaemolyticus* [31]. To determine whether read-through transcription of *vtrB* activates T3SS2-mediated cytotoxicity, human colon adenocarcinoma Caco-2 cells were infected with *V. parahaemolyticus* strain POR-2 or its derivative DT, and the level of cytotoxicity was evaluated (Fig 6A). The POR-2 strain caused complete cell death at 6 hours postinfection, whereas *vtrB* deletion from the POR-2 strain showed attenuated

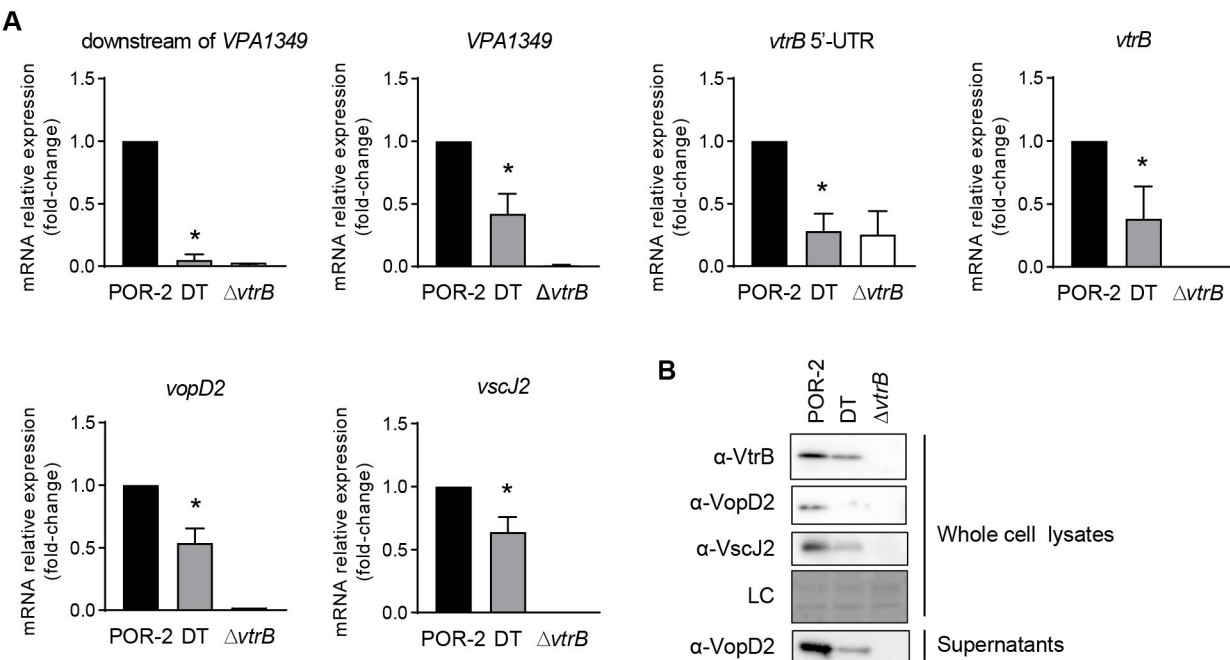

**Fig 5. Read-through transcription extending to *vtrB* is needed for activation of T3SS2 gene expression.** (**A**) Effect of the double terminator on the transcription read-through and expression of VtrB-regulated genes. *V. parahaemolyticus* POR-2, POR-2 DT (DT) and POR-2 Δ*vtrB* (Δ*vtrB*) strains were grown under the inductive condition, and total RNA was extracted from each culture once the culture reached an $OD_{600}$ of 1. Relative expression of downstream of *VPA1349* (between *VPA1349*T and the transcription start site of *vtrB*), *VPA1349*, *vtrB* 5'-UTR, *vtrB*, *vscJ2*, and *vopD2* with the housekeeping gene *recA* was analyzed by qRT–PCR. The values represent the means ± SDs from a minimum of three independent experiments. *, $p < 0.05$, compared with POR-2 by Student's t test. (**B**) Effect of the double terminator on the production of VtrB and T3SS2-related proteins. Bacterial whole-cell lysates and culture supernatants of *V. parahaemolyticus* POR-2, DT, and Δ*vtrB* strains grown under the inductive condition to an $OD_{600}$ of 1.8 were analyzed by immunoblotting with the indicated antibodies. Whole-cell lysate proteins on the blotted membrane were visualized with Ponceau S staining for a loading control (LC).

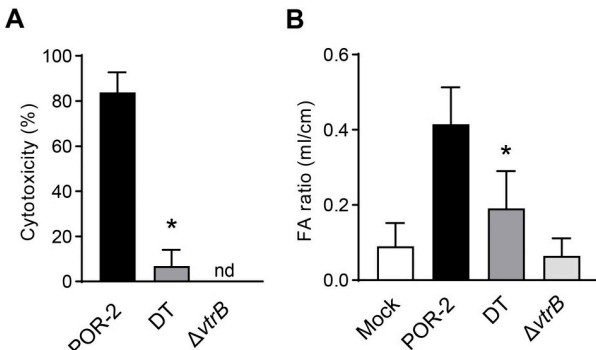

**Fig 6. Read-through transcription of *vtrB* is needed for the ability of *V. parahaemolyticus* to induce pathogenicity.** (**A**) Cytotoxicity was evaluated in Caco-2 cells infected with POR-2, DT, and Δ*vtrB* strains grown under the inductive condition prior to infection at a multiplicity of infection of 10. After 6 hours of infection, the cytotoxic activity was evaluated by determining amount of the lactate dehydrogenase release. The values are the means ± SDs (n = 4). nd, not detected; *, $p < 0.05$ compared with POR-2 by Student's t test. (**B**) Fluid accumulation in rabbit ileal loops infected with the indicated *V. parahaemolyticus* strains. Each ligated loop was infected with bacteria at $10^9$ colony-forming units or was not infected (mock), and the fluid accumulation in the loop was assessed 16 hours after infection. The FA ratio represents the amount of accumulated fluid (ml) per length (cm) of ligated rabbit small intestine. The values are the means ± SDs (n = 4). *, $p < 0.05$, compared with POR-2 by Student's t test.

cytotoxicity due to lack of T3SS2 gene activation. Notably, the DT strain exhibited markedly reduced cytotoxicity at 6 hours postinfection. The diarrhea-inducing activity of *V. parahaemolyticus* also depends on T3SS2 [5,31]. We evaluated the effect of read-through transcription of *vtrB* on diarrhea-inducing activity in the rabbit ileal loop model. The ligated ileal loops of rabbits were inoculated with *V. parahaemolyticus* strains, and at 16 h postinfection, the fluid accumulation in each loop was measured. The DT strain impaired fluid accumulation in rabbit ileal loops, compared with the parental POR-2 strain (Fig 6B). Taken together, these results support the notion that read-through transcription of *vtrB* mediates elevated T3SS2 gene expression and ensures the T3SS2-mediated pathogenicity of *V. parahaemolyticus*.

## Discussion

Bacteria have evolved transcription regulatory networks for the appropriate control of gene expression in response to environmental changes [32,33]. The autoregulation of transcriptional regulators is a common strategy consisting of regulatory networks for positive or negative feedback loops that amplify or reduce the output [18,34,35]. The typical mode of autoregulation of a transcription factor is achieved by acting on its own promoter and thereby enhancing or repressing its own gene transcription [17,18]. Indeed, the autoregulation of a transcription factor depending on its own promoter is often observed with master regulators of T3SS gene expression in gram-negative pathogens, such as *Salmonella* HilD [36,37], *Shigella flexneri* VirB [38], and enteropathogenic *E. coli* Ler [39]. In this study, we found that *V. parahaemolyticus* VtrB, the master regulator of T3SS2 gene regulation, also positively autoregulates its own gene expression, but not in a canonical manner dependent on its own proximal promoter. The emerging picture from this study is that the nascent VtrB resulting from the initial activation of the *vtrB* gene secondarily activates its own expression by generating heterogeneous transcripts from the distal promoter of the upstream operon that read through the intrinsic terminator, which connects the autoregulatory loop for *vtrB* gene expression and can lead to robust expression of the T3SS2 genes for virulence (Fig 7). Among multiple transcription units containing the *vtrB* gene, one (S transcript) was initiated from the *vtrB* promoter, and the others (L and M transcripts) were initiated with its 5'-end located further upstream of the *vtrB* promoter at the internal sites within the *VPA1356–VPA1349* operon, the transcription of which is activated by VtrB. An operon is generally defined as a transcription unit containing multiple genes with a single promoter positioned just upstream of the first gene in the operon, whereas operons are often split into suboperons by internal promoters [40–43]. However, because the L and M transcripts were expected to result from transcription initiated from the *VPA1356* promoter (L and M transcript precursor) (Fig 3B), the precursor is likely to undergo some modification to be truncated to these transcripts. Such modification may occur simultaneously with transcription, because *VPA1356–vtrB* (~7.5 k nt) and *VPA1356–VPA1349* (~6.7 k nt) transcripts were not detected in our northern blotting assay (Figs 2 and 4, S4C and S4D). Indeed, bacterial RNA undergoes many cleavage events, which are often operated by ribonucleases (RNases) [44]. Because the L and M transcript retain their 3'-end with the *vtrB* gene, the 3' to 5' exonucleases are likely not associated with the processing of the precursor of these transcripts. Moreover, given the absence of 5' to 3' exonucleases in γ-proteobacteria [45,46], the processing event is likely endonucleolytic cleavage catalyzed by endonucleases. Bacteria have a repertoire of endonucleases [45,47], and the RNase responsible for this processing needs to be addressed in a future study. In addition, the L transcript exhibited a gradual decline starting 30 min after TDC induction (Fig 2C), suggesting that the L transcript may undergo further truncation into the M transcript. The decline in the S transcript starting 30 min after TDC induction observed in the WT strain is a read-through transcription-dependent process

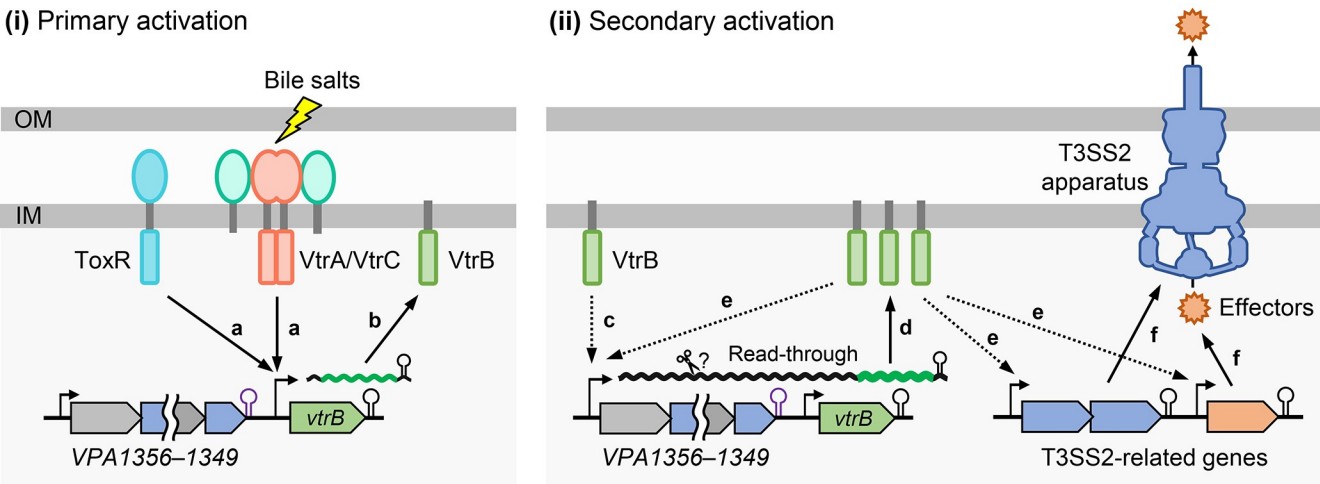

**Fig 7. A working model depicting an autoregulatory loop of *vtrB* expression connected by read-through transcription.** VtrA complexed with VtrC initiates activation of *vtrB* transcription at the *vtrB* promoter in cooperation with ToxR (**a**), which leads to primary expression of VtrB (**b**). VtrB then activates transcription from the promoter upstream of *VPA1356* (**c**), part of which reads through the intrinsic terminator downstream of *VPA1349*, resulting in an extended transcript that reaches *vtrB* (read-in transcription of *vtrB*) and increased VtrB expression (**d**). VtrB induces further activation of transcription from the promoter upstream of *VPA1356* and activates transcription of T3SS2-related genes (**e**), which can lead to robust expression of T3SS2 genes for virulence (**f**). OM, outer membrane; IM: inner membrane.

because it was not observed in the Δ*vtrB* and DT strains (Figs 2C and S4C), suggesting the possibility that transcription elongation from upstream might interfere with transcriptional activation by VtrA on the *vtrB* promoter. Alternatively, a potential factor encoded in the VtrB-regulated regulon may antagonize VtrA-activated *vtrB* transcription. These issues warrant further study.

Our data demonstrated read-through transcription across the intrinsic terminator located at the end of the *VPA1356*–*VPA1349* operon. Indeed, terminator read-through is a more widespread event than previously thought, as observed in *E. coli* [48] and *B. subtilis* [49]. The notion that *VPA1349*T with a computed $\Delta G_T$ value of −8.80 kcal/mol is a less stable terminator is consistent with the results previously obtained by Zhai, et al. who classified the 214 characterized terminators in *E. coli* into strong and weak terminators, with the strong terminators having a low $\Delta G_T$ of −17 to −14 kcal/mol [50]. A strong and stable hairpin is expected to cause efficient termination due to a greater ability to push the RNAP away from the DNA strand [51]. The lower stability of the hairpin formed by *VPA1349*T may explain why read-through transcription is able to occur over this terminator. Alternatively, the distance between the intrinsic terminator and the stop codon of the gene also affects the termination efficiency, which is reduced when the stop codon is only a few base pairs upstream of the terminator, in which case the translating ribosome may directly repress the hairpin folding of the terminator of the nascent mRNA, resulting in read-through transcription due to an insufficient termination capacity [52,53]. *VPA1349*T is separated from the stop codon of the *VPA1349* gene by only one A nucleotide (S3A Fig), suggesting the possibility that the proximity between the terminator and the stop codon of the *VPA1349* gene may affect the transcription read-through. Trans-acting factors also control the termination efficiency by directly regulating hairpin folding or by interfering with RNAP pausing at the termination site [54]. The involvement of trans-acting factors in transcription termination in *V. parahaemolyticus* is not yet evident, but these hypotheses regarding *VPA1349*T deserve future studies.

Our observation with the double terminator mutant indicates that much of the VtrB expression induced by bile acid stimulation is derived from autoactivation (Fig 5). This

autoactivation loop was found to be essential for enabling the T3SS2 gene expression response to exert pathogenicity, which suggests the necessity of this response for adaptation to the environment in the human gastrointestinal tract. Autoactivation theoretically involves a slower induction rate but increases the sensitivity of the circuit response to the input signal and amplifies the longitudinal output; this process is favored by the prescribed response [55]. This property may be advantageous for pathogenic bacteria in the stage of infection that lasts hours, in terms of increasing the sensitivity to host-derived signals and maintaining a virulence gene expression level that is sufficient for virulence. Indeed, positive autoregulation of a transcription factor is often seen with regulatory genes for virulent T3SS, such as HilD of *Salmonella enterica*, VirB of *Shigella flexneri*, and ExsA of *Pseudomonas aeruginosa* [38,56,57]. *Salmonella* HilD binds to its own promoter for self-activation [56], whereas *Shigella virB* is activated by the transcriptional activator VirF, and VirB then acts on its own promoter but also on the *virF* promoter to activate *virF* transcription, forming positive feedback loops [38]. In contrast to these canonical autoactivation mechanisms, *Pseudomonas aeruginosa exsA* is initially activated by Vfr in an *exsA* promoter ($P_{exsA}$)-dependent manner, and ExsA then binds to the promoter of the upstream *exsCEB* operon ($P_{exsC}$) to generate a polycistronic *exsCEBA* transcript; during this process, ExsA does not bind to or activate its own promoter [58]. The $P_{exsC}$ promoter is approximately 400-fold more active than $P_{exsA}$ and is thus expected to contribute to higher *exsA* transcript levels [58]. *exsA*-like autoactivation (transcribed as a distal promoter-dependent operon) is observed with another AraC family virulence regulator in gram-negative bacteria, RegA, the global virulence regulator of *Citrobacter rodentium* [59]. Thus, the case of *exsA* autoactivation appears to be similar to that of *vtrB* in terms of polycistronic transcription from the distal promoter rather than the proximal promoter, which may represent an efficient strategy to simultaneously activate its own gene and its target genes from a single promoter but can be considered a polymorphism of the operon structure composed of *exsCEBA* genes due to the lack of predicted intrinsic terminator downstream of *exsCEB*.

In terms of autoactivation via read-through transcription beyond the terminator, greater similarity is found in the regulation of the *toxT* gene encoding the transcription factor ToxT of *V. cholerae*, a causative agent of the severe diarrheal disease cholera, and a closely related species of *V. parahaemolyticus*. ToxT directly activates the *tcp* operon encoding the Tcp pilus needed for *V. cholerae* intestinal colonization [60,61], and the *ctxAB* gene encoding the cholera toxin, the absolute virulence factor of *V. cholerae* for causing severe watery diarrhea [60,62]. The activation of *toxT* transcription is initiated by binding of the membrane-bound transcription factor TcpP and its cotranscribed factor TcpH just upstream of the RNAP-binding site on the *toxT* promoter, which is promoted by binding of an additional membrane-bound transcription factor, ToxR, to the *toxT* promoter [63,64]. This regulatory pathway is quite similar to the initial activation of *vtrB* transcription: the VtrA/VtrC complex binds around the -35 element on the *vtrB* promoter, and the ToxR homolog binds further upstream, indicating functional homology between the two regulatory pathways without sequence homology. Notably, this transcription from the *tcpA* promoter reads through a relatively inefficient terminator downstream of *tcpF* (upstream of the *toxT* gene), resulting in the transcription extending to the *toxT* gene, which allows amplification of *toxT* gene expression and optimal production of Tcp pilus and cholera toxin [61,65]. Interestingly, *V. cholerae* and *V. parahaemolyticus* thus share not only functional homology in the virulence regulatory pathway but also the mechanism by which the virulence regulator is positively autoregulated, suggesting convergent evolution of virulence regulation in *Vibrio* species. Although it is difficult to discuss the utility of read-through transcription-mediated autoactivation for bacteria based only on these limited examples, one possible inference is that moderate transcription termination at the upstream operon terminator fine-tunes the rate of downstream read-through transcription, resulting in

sustained activation of the regulator and subsequent target genes. Because both Tcp pilus and T3SS2 are membrane-localized macromolecular apparatuses whose assembly is temporally sequential and hierarchical [66,67], gradual increases in protein amounts may facilitate accurate assembly. Alternatively, read-through transcription with moderate termination may optimize the amount of VtrB production for T3SS2 gene expression, which may be supported by the observation that a *V. parahaemolyticus* strain carrying *VPA1349*T-HP mutation on the chromosome, which is defective in transcription termination at the end of the upstream operon, showed an increased *vtrB* expression level but no significant change in T3SS2 gene expression compared with the parental strain (S5 Fig).

In summary, we propose that positive autoregulation of *V. parahaemolyticus* VtrB, which involves read-through transcription from the upstream operon in the *vtrB* gene, is essential for T3SS2 gene expression and virulence. Therefore, our study offers new insights into how *V. parahaemolyticus* controls virulence gene expression to ensure pathogenicity. A similar regulatory mechanism may be applied to any transcription factor with the following prerequisite for autoactivation by read-through transcription: located downstream of its target operon with a weak terminator in the same orientation as transcription.

## Materials and methods

### Ethics statement

All animal experiments in this study were conducted in strict accordance with the guidelines for the Care and Use of Laboratory Animals in the Research Institute for Microbial Diseases, Osaka University, and were performed following an experimental protocol approved by the Animal Care and Use Committee of the Research Institute for Microbial Diseases, Osaka University.

### Bacterial strains and plasmids

All bacterial strains and plasmids used in this study are listed in S2 and S3 Tables. The clinical isolate RIMD2210633 of *V. parahaemolyticus* was used as the wild-type strain (WT) in this study [4]. All *V. parahaemolyticus* strains were grown at 37˚C in Luria–Bertani (LB) medium with modified NaCl concentrations (tryptone, 1%; yeast extract, 0.5%; NaCl, 0.3 M). Appropriate antibiotics were added to grow the plasmid-carrying strains: 20 μg/ml gentamycin for pHRP309 and 15 μg/ml chloramphenicol for pBAD18-Cm plasmid backgrounds. Arabinose at a final concentration of 0.1% was added to the strains harboring pBAD18-Cm-based plasmids at the early exponential growth phase to induce $P_{BAD}$-driven expression. The primers used for plasmid construction are listed in S4 Table. *Escherichia coli* DH5α and BW19851 strains were used for the general manipulation of plasmids and the mobilization of plasmids into *V. parahaemolyticus*.

### Mutant construction

A four-primer PCR technique was used to construct insertion and deletion mutants as previously described [30]. The primers used are listed in S4 Table. Briefly, the DNA fragment of the deletion or insertion target was cloned and inserted into the pCRII-TOPO vector. This fragment was extracted from the pCRII-TOPO vector by digestion with the restriction enzymes BamHI and PstI, and then cloned into an R6K-ori suicide vector, pYAK1, which contains the *sacB* gene that confers sensitivity to sucrose. The resulting deletion or insertion plasmid was introduced into *E. coli* BW19851 and transferred into *V. parahaemolyticus* strains by conjugation. The resulting conjugates were selected on thiosulfate citrate bile sucrose agar containing

chloramphenicol at a concentration of 5 μg/ml and then screened for mutants on LB plates supplemented with 10% sucrose. The desired deletion/insertion in the *V. parahaemolyticus* genome was confirmed by PCR.

## RNA extraction

Total RNA was extracted using the hot acidic phenol RNA isolation method [68]. For qRT–PCR and northern blotting analysis, *V. parahaemolyticus* cells were grown in LB broth containing 0.3 M NaCl and 80 μM TDC at 37˚C until the optical density at 600 nm ($OD_{600}$) reached 1.0. For mapping the 5'-ends of transcripts, *V. parahaemolyticus* cells were grown in LB broth containing 0.3 M NaCl at 37˚C until the $OD_{600}$ reached 0.8 and then treated with 80 μM TDC for 15 min. The cultures were harvested by centrifugation at $3,000 \times g$ for 10 min, and the supernatant was removed. The remaining pellets were lysed with lysis buffer (0.5% SDS, 20 mM sodium acetate, and 10 mM EDTA at pH 5.5), followed by the addition of acid phenol. The mixture was then heated at 60˚C for 10 min. After centrifugation, the upper aqueous layer was collected. The RNA was precipitated with ethanol and dried until the resulting pellets became clear. The precipitated RNA was dissolved in nuclease-free water and treated with Turbo DNase (Thermo Fisher) according to the manufacturer's instructions. The concentration of RNA was measured using a NanoPhotometer spectrophotometer (Implen, USA).

## Quantitative real-time polymerase chain reaction (qRT–PCR)

qRT–PCR was performed as previously described [13], with a slight modification. Briefly, the purified RNA samples were diluted to 20 ng/μL with nuclease-free water, and the reactions were performed using SYBR Green RNA-direct Real-time PCR Master Mix (Toyobo) and a QuantStudio 5 real-time PCR system (Thermo Fisher), according to the manufacturer's instructions. Relative quantification was performed using the threshold cycle ($2^{-\Delta\Delta CT}$) method and was normalized to that of *recA* as the housekeeping gene.

## Transcriptional reporter assay

*V. parahaemolyticus* strains harboring pHRP309-derived *lacZ* reporter plasmids were grown in LB broth containing 0.3 M NaCl at 37˚C to an $OD_{600}$ of 0.8. The bacterial cultures were then supplemented with 80 μM TDC and further incubated until reaching an $OD_{600}$ of 1.8. For pBAD-derived reporter plasmids, *V. parahaemolyticus* strains were grown in LB broth containing 0.3 M NaCl with or without 80 μM TDC at 37˚C to an $OD_{600}$ of 1. Arabinose was added at the early exponential growth phase to a final concentration of 0.1%. The β-galactosidase activity of the bacterial cell lysates was measured using Miller's method with the substrate *o*-nitrophenyl-β-D-galactopyranoside (ONPG), as described previously [69].

## Mapping of 5'-end of the transcript

The 5'-ends of the transcripts (cDNA fragments) were determined using the SMARTer RACE 5'/3' Kit (Takara Bio, Shiga, Japan) according to the manufacturer's instructions. Total RNA from *V. parahaemolyticus* was isolated as described above. Sanger DNA sequencing was outsourced to Genewiz (Azenta Life Science). The sequence information obtained was analyzed using GENETYX ver. 18.0.4 software (GENETYX, Tokyo, Japan).

## Northern blot analysis

Extracted total RNA (2.5 μg) was heated at 60˚C and subjected to 1.2% agarose gel electrophoresis in the presence of formaldehyde, with Dyna Marker (Prestain Marker for RNA High,

BioDynamics Laboratory Inc., Japan). After electrophoresis, the RNA was transferred from the agarose gel to a positively charged nylon membrane (Roche, Germany) overnight with 20× SSC buffer (3 M NaCl, 0.3 M sodium citrate dehydrate) and cross-linked by UV irradiation $(1200 \times 100 \ \mu J/cm^2)$. Hybridization was performed at 50˚C for 4 h in DIG Easy Hyb buffer (Roche, Germany) with a DIG-labeled probe. DIG-labeled probes were amplified using a PCR DIG probe synthesis kit (Roche, Germany). The primers used for probe synthesis are listed in S4 Table. The membrane was rinsed with a high-stringency buffer (0.1× SSC buffer, 0.1% SDS) and incubated with a blocking solution (Roche, Germany) for 30 min. Anti-digoxigenin-alkaline phosphatase (anti-DIG-AP) (Roche, Germany) was added to the blocking buffer-soaked membrane. The membrane was incubated for 1 h, equilibrated with detection buffer (0.1 M Tris-HCl at pH 9.5, 0.1 M NaCl), developed with the chemiluminescent substrate CDP-STAR (Roche, Germany) and visualized using an Amersham ImageQuant 800 system (Cytiva, USA). Ribosomal RNA was stained with Gel Red (Biotium, USA) and visualized by UV irradiation as a loading control.

## Protein sample preparation

*V. parahaemolyticus* strains were grown in LB broth containing 0.3 M NaCl and 80 μM TDC at 37˚C for 3 h to an $OD_{600}$ of 1.8. The cultures were centrifuged at $15,000 \times g$ for 2 min to separate the supernatants and the resulting pellets, which were used as whole-cell lysates. The secreted proteins were precipitated from the supernatants with ice-cold trichloroacetic acid at a final concentration of 10% on ice for 1 h and then centrifuged at $15,000 \times g$ for 30 min at 4˚C. The resulting pellets were washed with cold acetone and centrifuged at $15,000 \times g$ for 30 min at 4˚C. The precipitate was dried at room temperature for 20 min. Proteins of whole cell lysates and supernatants were solubilized in Laemmli buffer, sonicated for 5 min, and denatured at 95˚C for 5 min. The samples were then subjected to SDS–PAGE and immunoblot analysis.

## Immunoblot analysis

For immunoblot analysis, protein samples were separated by SDS–PAGE and transferred to a PVDF membrane by semidry electroblotting. The membranes were blocked in TBST (20 mM Tris-HCl at pH 7.4, 150 mM NaCl, and 0.1% Tween 20) containing 5% skim milk for 1 h and then incubated overnight with primary antibodies against the protein of interest. The anti-VtrB, anti-VscJ2, and anti-VopD2 polyclonal antibodies were prepared in-house by immunizing New Zealand White rabbits, as described elsewhere [11,30]. The membranes were then probed with horseradish peroxidase-conjugated goat anti-rabbit antibody (code: 62–1820, Invitrogen) for 2 h at room temperature. The blots were then developed using an ECL Prime Western Blotting Detection Reagent (Cytiva, USA) and visualized using the Amersham Image-Quant 800 system (Cytiva, USA). For the loading control, whole cell lysates on the blotted membrane were stained with Ponceau S staining solution (Thermo Fisher) for visualization before the blocking procedure.

## Cell culture and cytotoxicity assay

Caco-2 cells (ECACC 86010202) supplied by the European Collection of Authenticated Cell Cultures were maintained in Dulbecco's modified Eagle's medium (DMEM) supplemented with 10% fetal bovine serum (FBS) and 100 μg/ml gentamicin at 37˚C with 5% $CO_2$. The cytotoxicity assay was performed as previously described [31,70], with some modifications. Briefly, $2 \times 10^4$ Caco-2 cells were seeded into each well of a 96-well plate and grown to confluence for 48 hours. The cells were washed twice with phosphate-buffered saline, and the medium was

then replaced with fresh phenol red-free DMEM. Prior to infection, *V. parahaemolyticus* strains grown in LB broth containing 0.3 M NaCl and 80 μM TDC for 3 hours were harvested by centrifugation and suspended in PBS. The cells were then infected with each *V. parahaemolyticus* strain at a multiplicity of infection (MOI) of 10 for 6 hours, and the cytotoxicity assay was performed by measuring the release of lactate dehydrogenase (LDH) into the culture supernatants with a Cytotoxicity LDH assay kit-WST (Dojindo) according to the manufacturer's instructions. The cytotoxicity percentage was calculated with the following equation: [optical density at 490 nm ($OD_{490}$) of experimental release–$OD_{490}$ of spontaneous release] / [$OD_{490}$ of maximum release–$OD_{490}$ of spontaneous release] × 100. Spontaneous release refers to the amount of LDH released from uninfected cells, whereas maximum release is the total amount of LDH released after the complete lysis of uninfected cells by detergent.

## Rabbit ileal loop assay

Rabbit ileal loop tests were performed as previously described [70]. Briefly, 1 ml of the bacterial suspensions ($10^9$ colony-forming units) was injected into the ligated ileal loops of a 1.5-kg female New Zealand White rabbit (the length of a loop is approximately 8 cm), and the fluid accumulation in each loop was measured 16 h after inoculation. The fluid accumulation (FA) ratio represents the amount of accumulated fluid (ml) per length of ligated rabbit small intestine (cm).

## Statistical analysis

GraphPad Prism 9.5.1 software was used for the statistical analysis of all the data. A two-tailed Student's t test or one-way ANOVA followed by Dunnett's multiple comparison test was used for the statistical analysis, and p values $< 0.05$ were considered to indicate statistical significance.

## Supporting information

**S1 Fig.** Schematic representation of the vtrB upstream region with adjacent genes in the WT (A) and Δ*vtrB* strains (B). The arrows indicate genes with their orientation. The nucleotide position is based on the transcriptional start site of *vtrB* (indicated as +1). The coding sequence of *vtrB* has a length of 552 bp, whereas the Δ*vtrB* strain contains a 410-bp deletion in the coding sequence.
(TIF)

**S2 Fig. Mapping of the 5'-ends of *vtrB* transcripts.** (**A**) Agarose gel electrophoresis of 5'-RACE PCR products. (**B**) Schematic representation of the position of the mapped 5'-end of each *vtrB* transcript from 5'-RACE PCR. (**C**) Nucleotide sequence around the 5'-ends of 5'-RACE products. The determined 5'-ends are indicated in red. The transcriptional start site of the *vtrB* is indicated as +1, and putative −35 and −10 elements are underlined. The shading indicates the region of coding sequences, and the names of the genes are shown above.
(TIF)

**S3 Fig. Transcription termination ability of *VPA1349*T.** (**A**) The secondary structure of the *VPA1349* terminator (*VPA1349*T) and the *E. coli rplL* terminator (*rplL*T) predicted using Mfold [22]. Gray shading indicates the substituted nucleotides (GGGGC > CCCCG) for disrupting the hairpin structure formation in *VPA1349*T. (**B**) Evaluation of the transcription termination ability using the terminator-fused *lacZ* reporters. The *V. parahaemolyticus* Δ*vtrB* strain harboring each reporter plasmid with *VPA1349*T, *rplL*T, or the *VPA1349* terminator hairpin mutant (*VPA1349*T-HP) was grown in LB medium containing 0.3 M NaCl with or

without TDC induction, and the β-galactosidase activity was monitored. The values show the means and error bars represent the SDs (n = 3). nd, not detected; *, $p < 0.05$, as revealed by one-way ANOVA followed by Dunnett's multiple comparison test. (**C**) Schematic structure of the general intrinsic terminator (adapted from [50] with modifications).
(TIF)

**S4 Fig. Effective transcription termination by the double terminator system.** (**A**) The secondary structure of the double terminator system composed of *VPA1349*T and *rplL*T was predicted using Mfold [22]. (**B**) Transcription termination ability of the double terminator system. *V. parahaemolyticus* Δ*vtrB* with *lacZ* reporter plasmids containing the *VPA1349* gene with *VPA1349*T or with the double terminator (*VPA1349*-DT) was grown in LB medium containing 0.3 M NaCl with or without TDC, and the β-galactosidase activity was evaluated. The values show the means and error bars represent the SDs (n = 3). nd, not detected; *, $p < 0.05$, compared with *VPA1349*-T by Student's t test. (**C**) Effect of the double terminator on the *vtrB* transcript profile in *V. parahaemolyticus*. The WT and DT strains were grown to an $OD_{600}$ of 0.8, and TDC was then added. RNA was extracted after 0, 5, 15, 30, 45, and 60 min of TDC induction, and northern blotting was performed using the $P_{vtrB}$ probe. L, L transcript; M, M transcript; S, S transcript. 23S rRNA and 16S rRNA served as loading controls. (**D**) Transcription termination at the double terminator. *V. parahaemolyticus* WT and DT strains were grown in LB medium containing 0.3 M NaCl to an $OD_{600}$ of 0.8, and TDC was then added. After further incubation for 15 min, total RNA was extracted, and northern blotting was performed using the indicated probes: the *VPA1350* coding region (left), the *VPA1349* coding region (center) and $P_{vtrB}$ (right). L, L transcript; LT, L transcript terminated at *VPA1349*T or the double terminator; M, M transcript; MT, M transcript terminated at *VPA1349*T or the double terminator; S, S transcript. The data are representative of three independent experiments (**C**, **D**).
(TIF)

**S5 Fig. Disrupting the hairpin structure of *VPA1349*T in the chromosome increases *vtrB* expression but not VtrB-regulated gene expression.** (**A**) *V. parahaemolyticus* strains WT, WT-derived DT (DT), Δ*vtrB*, and a strain carrying the hairpin mutation in *VP1349*T (S3A Fig: *VP1349*T-HP) on the chromosome of the WT strain (HP strain) were grown in LB medium containing 0.3 M NaCl to an $OD_{600}$ of 0.8, and TDC was then added. After further incubation for 15 min, total RNA was extracted, and northern blotting was performed using the probe for the *VPA1349* coding region. The HP strain was defective in transcription termination at the end of the upstream operon, as observed by the absence of L and M transcripts terminated at *VP1349*T. L, L transcript; LT, L transcript terminated at *VPA1349*T; M, M transcript; MT, M transcript terminated at *VPA1349*T. The data are representative of three independent experiments. (**B**) Effect of the hairpin mutation in *VPA1349*T on *vtrB* and VtrB-regulated gene expression. *V. parahaemolyticus* strains POR-2, POR-2 DT (DT), POR-2 HP (HP), and POR-2 Δ*vtrB* (Δ*vtrB*) were grown under the inductive condition, and total RNA was extracted from each culture once the culture reached an $OD_{600}$ of 1. Relative expression of *vtrB*, *vopD2*, and *vscJ2* with the housekeeping gene *recA* was analyzed by qRT–PCR. The values represent the means ± SDs from a minimum of three independent experiments. *, $p < 0.05$; ns, not significant, compared with POR-2 by Student's t test. (**C**) Effect of the hairpin mutation in *VPA1349*T on the production of VtrB and T3SS2-related proteins. Bacterial whole-cell lysates and culture supernatants of *V. parahaemolyticus* POR-2, HP, and Δ*vtrB* strains grown under the inductive condition to an $OD_{600}$ of 1.8 were analyzed by immunoblotting with the indicated antibodies. Whole-cell lysate proteins on the blotted membrane were visualized with

Ponceau S staining for loading control (LC).
(TIF)

**S1 Table. Gibbs free energy (ΔG) values of *VPA1349*T and *rplL*T, calculated using Mfold** [22].
(PDF)

**S2 Table. Bacterial strains used in this study.**
(PDF)

**S3 Table. Plasmids used in this study.**
(PDF)

**S4 Table. Primers used in this study.**
(PDF)

**S1 Data. Source data for graphs in this study.**
(XLSX)

## Acknowledgments

We thank the members of the Department of Bacterial Infections for helpful discussions.

## Author Contributions

**Conceptualization:** Dhira Saraswati Anggramukti, Eiji Ishii, Shigeaki Matsuda.

**Funding acquisition:** Eiji Ishii, Shigeaki Matsuda.

**Investigation:** Dhira Saraswati Anggramukti, Eiji Ishii, Shigeaki Matsuda.

**Project administration:** Eiji Ishii, Shigeaki Matsuda.

**Resources:** Andre Pratama, Mohamad Al Kadi, Toshio Kodama.

**Supervision:** Tetsuya Iida, Toshio Kodama, Shigeaki Matsuda.

**Validation:** Eiji Ishii.

**Visualization:** Dhira Saraswati Anggramukti, Eiji Ishii.

**Writing – original draft:** Dhira Saraswati Anggramukti, Shigeaki Matsuda.

**Writing – review & editing:** Eiji Ishii, Shigeaki Matsuda.

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
