## [Decision Letter · Decision Letter 0]

26 Jan 2024

Dear Dr. Matsuda,

Thank you very much for submitting your manuscript "The read-through transcription-mediated autoactivation circuit for virulence regulator expression drives robust type III secretion system 2 expression in Vibrio parahaemolyticus" for consideration at PLOS Pathogens. As with all papers reviewed by the journal, your manuscript was reviewed by members of the editorial board and by several independent reviewers. The reviewers appreciated the attention to an important topic. Based on the reviews, we are likely to accept this manuscript for publication, providing that you modify the manuscript according to the review recommendations.

Sincerely,

Matthew C Wolfgang

Section Editor

PLOS Pathogens

Matthew Wolfgang

Section Editor

PLOS Pathogens

Michael Malim

Editor-in-Chief

PLOS Pathogens

orcid.org/0000-0002-7699-2064

Reviewer Comments (if any, and for reference):

Reviewer's Responses to Questions

**Part I - Summary**

Reviewer #1: In this study, the authors demonstrated that autoactivation of VtrB is induced by transcription from the operon upstream of vtrB through read-through transcription across the intrinsic terminator, which enlarged the expression of T3SS2 and the virulence of V. parahaemolyticus. The paper was written clearly and easy to understand.

I have only one comment, that is, in Figure 1d, the author only investigated the effect of overexpression of vtrA or vtrB on the promoter activity of vtrB in the ΔvtrAΔvtrB strain. The author should simultaneously test the promoter activity of vtrB in WT, ΔvtrA and ΔvtrB.

Reviewer #2: The authors present a paper with novel findings of vtrB regulation in V. parahaemolyticus. The authors present evidence ΔvtrB strains are deficient in expressing the 5’-UTR of the vtrB gene. This goes against the prevailing model where VtrAC transcriptionally activates vtrB and the VtrB protein regulates the expression of the type three secretion system. They then go on to show that wild type cells express long and medium (4000 and 2000 base) transcripts containing the vtrB promoter, which is much longer than a 500 base transcript that is produced when transcription starts from the vtrB promoter. They then go on to show the promoter that is driving expression of these constructs is 7000 bp upstream of vtrB in the genome. The long and medium transcripts when probed for the genes upstream of vtrB. This probe revealed RNA which appear to be terminated before reaching the vtrB gene and they go on to identify a transcriptional terminator between VPI1349 and vtrB (VPI1349T). The authors test the transcriptional terminator strength by inserting it into the pBAD plasmid before lacZ. They compare the activity of the terminator to a known strong transcriptional terminator and a mutant VPI1349T. The VPI1349T showed repression of β-galactosidase activity that was not as complete as the strong terminator but significantly less than mutant VPI1349T. They then demonstrate increasing the terminator strength in V. parahaemolyticus negatively affects pathogenesis through gene expression of T3SS2 components, as well as with cytotoxic and animal models. The authors then present a novel model of vtrB regulation where VtrAC are responsible for starting transcription of vtrB, then VtrB takes over to autoregulate its gene.

Overall the authors have very thoughtfully laid out the evidence for a novel pathway of vtrB regulation. This work will be impactful as this model of transcriptional activity will be useful to the microbiology community at large. While I recommend the paper for publication there are some issues that need to be addressed.

**Part II – Major Issues: Key Experiments Required for Acceptance**

Reviewer #1: (No Response)

Reviewer #2: The authors address how having a strong terminator between VPI1349 and vtrB decreases virulence gene expression and pathogenesis. However, this does not address the physiological role of the wild type terminator. It would be nice to see if disrupting the terminator leads to non-terminated transcripts, altered gene expression, as well as cytotoxicity.

**Part III – Minor Issues: Editorial and Data Presentation Modifications**

Reviewer #1: (No Response)

Reviewer #2: Line 61: Replace the word “pathway” with machine or apparatus. The assembled secretion system is not a pathway.

Line 376: Says “… the increase in T3SS2 expression induced by vtrB autoactivation …” given that autoactivation occurs in the wildtype strains remove the words “the increase in”. In this section the authors are really checking whether decreasing terminator read through reduces vtrB expression and therefore markers of pathogenicity.

Line 401: Check units for rabbit ileal loop assay. Legend says “…accumulated fluid (ml) per length (cm)…” which I take to mean ml/cm while the y-axis of the graph has cm/ml.

Line 467: The authors are discussing read through transcription but mention the translating ribosome. I am guessing this should have read RNA polymerase.

PLOS authors have the option to publish the peer review history of their article (what does this mean?). If published, this will include your full peer review and any attached files.

Reviewer #1: No

Reviewer #2: No

Figure Files:

Data Requirements:

Reproducibility:

References:

---

## [Decision Letter · Decision Letter 1]

4 Mar 2024

Dear Dr. Matsuda,

We are pleased to inform you that your manuscript 'The read-through transcription-mediated autoactivation circuit for virulence regulator expression drives robust type III secretion system 2 expression in Vibrio parahaemolyticus' has been provisionally accepted for publication in PLOS Pathogens.

Best regards,

Matthew C Wolfgang

Section Editor

PLOS Pathogens

Matthew Wolfgang

Section Editor

PLOS Pathogens

Michael Malim

Editor-in-Chief

PLOS Pathogens

orcid.org/0000-0002-7699-2064

Reviewer Comments (if any, and for reference):

Reviewer's Responses to Questions

**Part I - Summary**

Reviewer #2: Overview:

The authors present a paper with novel findings of vtrB regulation in V. parahaemolyticus. The authors present evidence ΔvtrB strains are deficient in expressing the 5’-UTR of the vtrB gene. This goes against the prevailing model where VtrAC transcriptionally activates vtrB and the VtrB protein regulates the expression of the type three secretion system. They then go on to show that wild type cells express long and medium (4000 and 2000 base) transcripts containing the vtrB promoter, which is much longer than a 500 base transcript that is produced when transcription starts from the vtrB promoter. They then go on to show the promoter that is driving expression of these constructs is 7000 bp upstream of vtrB in the genome. The long and medium transcripts when probed for the genes upstream of vtrB. This probe revealed RNA which appear to be terminated before reaching the vtrB gene and they go on to identify a transcriptional terminator between VPI1349 and vtrB (VPI1349T). The authors test the transcriptional terminator strength by inserting it into the pBAD plasmid before lacZ. They compare the activity of the terminator to a known strong transcriptional terminator and a mutant VPI1349T. The VPI1349T showed repression of β-galactosidase activity that was not as complete as the strong terminator but significantly less than mutant VPI1349T. They then demonstrate increasing the terminator strength in V. parahaemolyticus negatively affects pathogenesis through gene expression of T3SS2 components, as well as with cytotoxic and animal models. The authors then present a novel model of vtrB regulation where VtrAC are responsible for starting transcription of vtrB, then VtrB takes over to autoregulate its gene.

Overall review:

Overall the authors have very thoughtfully laid out the evidence for a novel pathway of vtrB regulation. This work will be impactful as this model of transcriptional activity will be useful to the microbiology community at large. The reviewers have satisfactorily addressed all of my comments.

**Part II – Major Issues: Key Experiments Required for Acceptance**

Reviewer #2: none

**Part III – Minor Issues: Editorial and Data Presentation Modifications**

Reviewer #2: In the version I am looking at the Fig6B y axis still says cm/ml while the figure legend suggests it should read ml/cm. So please correct the figure or legend text.

PLOS authors have the option to publish the peer review history of their article (what does this mean?). If published, this will include your full peer review and any attached files.

Reviewer #2: No

---

## [Editor Report · Acceptance letter]

13 Mar 2024

Dear Dr. Matsuda,

We are delighted to inform you that your manuscript, "The read-through transcription-mediated autoactivation circuit for virulence regulator expression drives robust type III secretion system 2 expression in <i>Vibrio parahaemolyticus<i>," has been formally accepted for publication in PLOS Pathogens.

Best regards,

Michael Malim

Editor-in-Chief

PLOS Pathogens

orcid.org/0000-0002-7699-2064